# Physicochemical Characteristics, Antioxidant Properties and Consumer Acceptance of Greek Yogurt Fortified with Apple Pomace Syrup

**DOI:** 10.3390/foods12091856

**Published:** 2023-04-29

**Authors:** Jisoo Kim, Moonsook Kim, Ilsook Choi

**Affiliations:** 1Department of Food and Nutrition, Wonkwang University, Iksandae-ro, Iksan 54538, Republic of Korea; 2Department of Food and Nutrition, Wonkwang Health Science University, 514, Iksandae-ro, Iksan 54538, Republic of Korea; kmsook95@wu.ac.kr

**Keywords:** apple pomace, apple pomace syrup, Greek yogurt, antioxidant activity, consumer acceptance, functional food

## Abstract

Despite having high polyphenolic phytochemicals and functional components, apple pomace (AP) is often discarded in landfills, leading to pollution. The study aimed to find a sustainable application for AP in Greek yogurt fortified with AP syrup (APS). Physicochemical characteristics and antioxidant properties were analyzed for APS (APS0.00, APS1.25, APS2.50, APS3.75, APS5.00). As the AP content in the syrup increased, moisture content, titratable acidity, and viscosity significantly increased (*p* < 0.05). The total polyphenols and flavonoid content of APS increased with increasing AP content. In Greek yogurt fortified with APS (APY), reducing sugar content (0.55 mg/mL to 0.71 mg/mL) significantly increased with fermentation time and AP content, whereas pH level (6.85 to 4.28) decreased. The antioxidant activities by DPPH radical scavenging activity, ABTS radical scavenging activity, ferric reducing antioxidant power, and reducing power were also significantly increased with the AP content and fermentation time. In the consumer acceptance test of APY, APY1.25 had significantly high scores in overall acceptance, taste acceptance, and aftertaste acceptance with purchase intent (*p* < 0.05). The Greek yogurt fortified with APS as functional food had improved antioxidant properties and consumer acceptance, suggesting the possibility of developing sustainable AP products.

## 1. Introduction

Apple pomace (AP), which consists of apple peel and pulp remains, is a byproduct generated while processing apple juice and beverages. AP accounts for 20 to 30% of an apple’s weight and contains pectin substances, simple carbohydrates, protein, minerals, and polyphenols [1,2,3,4,5,6,7]. The polyphenols, including chlorogenic acid, catechin, epicatechin, procyanidin, and phloridzin in AP have functional characteristics such as antioxidant and antifatigue activity, and biological properties such as reducing the risk of cardiovascular disease, decreasing cholesterol level, reducing the risk of lung and colorectal cancer, and reducing the risk of type II diabetes, and weight loss [8,9,10].

The content of the polyphenols in AP is higher than that of fresh apples or juiced apples [11]. The functional component in AP varies depending on the extraction and drying process of the AP [12,13,14,15,16,17,18,19]. Currently, some AP is used as feed in livestock farms and organic fertilizer in orchards and greenhouses, but most are discarded [20]. A common method of disposal of AP is to dispose of it directly into the soil of landfills. However, it has been reported that the high moisture content of <70% and the abundance of biodegradable organic matter increase the risk of environmental pollution and public health when disposed directly into the soil [21]. It is considered necessary to find a professional waste treatment and utilization method of AP [6,22,23]. Such applications for AP have been actively researched by the industry due to AP’s high nutrition and favorable utilization [2,6,7,21]. Product development research using AP includes cookies [24], muffins [25], biscuits [26], baked products [27,28], drinks [29], fermented products [30], yogurts, and other products [31,32,33,34,35].

Yogurt contains probiotics, which promote human immunity and protect against several diseases with modulation of gut microbiota [36]. It promotes the absorption of minerals, vitamins, and lactose digestion, improves immunity, reduces blood cholesterol, and has anticancer effects [37,38,39,40,41]. It has been reported that yogurt is effective in preventing metabolic diseases such as diabetes, hyperlipidemia, and intestinal health [38,42]. The increased health consciousness among consumers has led to increased Greek yogurt consumption. Greek yogurt or concentrated yogurt is a fermented semi-solid and strained yogurt, which requires increased protein content before or after fermentation to a minimum of 5.6% compared with regular yogurt of a minimum of 2.7% [43]. Apple pomace syrup (APS) was prepared by apple pomace (AP) powder in five different addition concentrations (0.00%, 1.25%, 2.50%, 3.25%, and 5.00%), and physicochemical characteristics and antioxidant properties were evaluated. In this study, Greek yogurt was manufactured using APS (APS0.00, APS1.25, APS2.50, APS3.75, APS5.00) instead of apple pomace flour to avoid the precipitation of apple pomace flours but also promote the colloidal status during the fermentation process. Physicochemical characteristics and antioxidant properties of Greek yogurt fortified with APS (APY) were analyzed during fermentation periods to understand the effect of APS fortification on Greek yogurt. The consumer acceptance test with APY was conducted in the sensory booth to find a positive application for AP by developing functional yogurt fortified with APS.

## 2. Materials and Methods

### 2.1. Materials

The apples, *Malus pumila*, cultivated in Cheongsong-gun, Gyeongsangbuk-do, South Korea, were purchased online. Fructo-oligosaccharide (Cheiljedang, Incheon, Korea), fat-free milk containing protein 3 g/100 g (Maeil, Gwangju, Korea), skim milk powder containing protein 37 g/100 g (Seoul Milk, Yangju, Korea) and Greek yogurt (Bio Greek Yogurt, Maeil, Cheongyang, Korea) were purchased in Iksan. YoFlex Mild 1.0 (Chr. Hansen, Hoersholm, Denmark), a frozen culture composed of *Lactobacillus bulgaricus* and *Streptococcus thermophilus*, was purchased online and stored in a −18 °C freezer until use. These starter cultures were selected following the yogurt production under the US FDA Code of Regulation [44,45].

### 2.2. Preparation of Greek Yogurt Enhanced with Apple Pomace Syrup

The preparation process of Greek yogurt enhanced with APS is presented in Figure 1. First, the apples were immersed in a sodium hypochlorite solution (100 ppm concentration) for 5 min and then rinsed with tap water three times in accordance with the Korea Food Sanitation Act and School meal hygiene management guidelines [46]. The juice and the pomace were separated using a juicer (H-200, Hurom, Seoul, Korea). The pomace, with an initial moisture content of 81.62%, was dehydrated in a convection oven (OF-22GW, Jeio Tech, Siheung, Korea) at 60 °C for 24 h. The dried AP was ground at 800 rpm for 1 min using a knife mill (Knife mill, Fritsch, Idar-oberstein, Germany), and the AP powder was filtered through a 90 mesh. The final moisture contents of the dried AP powder were 10.46%.

The APS was prepared by adding the dried AP powder of different concentrations (0.00%, 1.25%, 2.50%, 3.75%, 5.00%) to the oligosaccharide solution (60%) and deionized water to make a total of 100%. The APS (APS0.00, APS1.25, APS2.50, APS3.75, APS5.00) was heated at 80 °C using a slow cooker (33130GL, Hamilton Beach, Glen Allen, VA, USA) until the total soluble solids reached 60 °Brix. The APS samples were stored in a refrigerator at 4 °C.

The APY was prepared by mixing fat-free milk (80 g, 2.40 g of protein), skim milk powder (12 g, 4.44 g of protein), frozen culture (0.01 g), and the APS (8.0 g) of different APS concentrations (APS0.00, APS1.25, APS2.50, APS3.75, APS5.00). The treatments were stirred at 330 rpm for 20 min, heated at 80 °C for 20 min, cooled to 40 °C while stirring, and then inoculated with a strain, stirred at 330 rpm for 20 min. Stirred solution of 30 g was dispensed into yogurt trays (7 cm × 3 cm) and fermented at 43 °C for 0, 3, 6, 8, and 12 h in an incubator (C-IND3, Changshin Science, Seoul, Korea).

### 2.3. Physicochemical Properties

The moisture content and crude ash content of APS and APY were determined in accordance with the techniques described by the official AOAC method (1990) [47]. The pH was measured with a pH meter (Seven Compact s220-k, Mettler Toledo, Greifensee, Switzerland) by taking 5 mL of APS and APY. The titrated acidity was diluted 10-fold by mixing 45 mL of distilled water with 5 mL of supernatant of the samples. The supernatant was obtained from APS and APY by centrifugation at 4000 rpm for 30 min (Combi 524R, Hanil, Daejeon, Korea) and titrated with 0.1 N NaOH solution until the pH reached 8.4. The viscosity of the samples was measured using a viscometer (Brookfield Digital Viscometer, Model DV III+, Brookfield Engineering Laboratories Inc., Stoughton, MA, USA) with an RV6 spindle at 20 rpm for 2 min at 25 °C. The colorimetric evaluation of the samples was measured using a colorimeter (CR-10 Plus, Konica Minolta, Tokyo, Japan) to obtain the colorimetric parameters, lightness (L*), green–red chromaticity coordinate (a*), and blue–yellow chromaticity coordinate (b*). The color differences (ΔE*) between the control and samples were calculated with the equation:$${\Delta E}^{*}={\left[{\left({\Delta L}^{*}\right)}^{2}+{\left({\Delta a}^{*}\right)}^{2}{\left({\Delta b}^{*}\right)}^{2}\right]}^{\left(1/2\right)}$$

A white calibration plate with the values L = 91.55, a = −0.32, and b = 0.28 was used to calibrate the apparatus before the measurements were carried out.

### 2.4. Antioxidant Component Analysis and Antioxidant Activity Assay

The total polyphenol content was analyzed using a method described by Arnous et al. (2001) [48] with minor modifications. APS and APY were centrifuged at 16,000 rpm for 30 min (Combi 524R, Hanil, Daejeon, Korea). Then, 50 μL Folin-Ciocalteu’s reagent was added to the supernatant of the sample (200 μL) and reacted for 5 min, and then 1000 μL of 2% Na2CO3 was added and reacted for 30 min. The absorbance was measured at 750 nm using an absorbance spectrometer (UN-1800, Shimadzu, Kyoto, Japan). The measured absorbance content was calculated from a standard curve using the standard material gallic acid (Sigma Aldrich, Co., St. Louis, MO, USA).

The total flavonoid content was analyzed using a method described by Shen et al. (2009) [49] with minor modifications. APS and APY were centrifuged at 16,000 rpm for 30 min (Combi 524R, Hanil, Daejeon, Korea). Then, 75 μL of 5% NaNO2 was added to the supernatant of the sample (200 μL) and reacted for 5 min, and then 150 μL of 10% AlCl3∙6H2O was added and reacted for 6 min. Then, 500 μL of 1 M NaOH was added and reacted in the dark for 11 min, and the absorbance was measured at 510 nm using an absorbance spectrometer (UN-1800, Shimadzu, Kyoto, Japan). The measured absorbance content was calculated from a standard curve using the standard material rutin (Sigma Aldrich, Co., St. Louis, MO, USA).

In antioxidant activity, the DPPH radical scavenging ability was analyzed using the method described by Garavand et al. (1958) [50] with minor modifications. For the DPPH reagent, 0.2 mM DPPH was prepared using methyl alcohol, and the absorbance value at 517 nm was 1.00 using an absorbance spectrophotometer (UN-1800, Shimadzu, Kyoto, Japan). Apple pomace syrup and Greek yogurt were centrifuged at 16,000 rpm for 30 min (Combi 524R, Hanil, Daejeon, Korea), 0.2 mM DPPH reagent 1 mL was added in the supernatant of the sample (200 μL) and reacted for 30 min in the dark. An absorbance spectrophotometer (UN-1800, Shimadzu, Kyoto, Japan) measured the absorbance at 517 nm and showed the scavenging ability (%).

ABTS (2,2′-azino bis-3-ethylbenzothiazoline-6-sulfonic acid) radical scavenging ability was analyzed by partially modifying the method by Re et al. (1999) [51]. For the ABTS reagent, 7 mM ABTS and 2.4 mM Potassium persulfate were mixed in equal amounts and reacted in the dark for 12 h to form ABTS+ (ABTS cation radical). Thereafter, a PBS solution was added so that the absorbance value was 0.70 using a 735 nm absorbance spectrometer (UN-1800, Shimadzu, Kyoto, Japan). Apple pomace syrup and Greek yogurt were centrifuged at 16,000 rpm for 30 min (Combi 524R, Hanil, Daejeon, Korea), 1 mL of ABTS+ solution was added in the supernatant of the sample (50 μL) and reacted in the dark for 30 min. The absorbance was measured at 735 nm using an absorbance spectrometer (UN-1800, Shimadzu, Kyoto, Japan). The scavenging ability was calculated from the standard curve using a standard material Trolox (Sigma-Aldrich, Co., St. Louis, MO, USA).

The Ferric Reducing Antioxidant Power assay was carried out according to the method by Benzie and Strain (1996) [52]. First, 0.2 M sodium acetate buffer, 10 mM TPTZ, 20 mM ferric chloride, and distilled water were mixed in a ratio of 10:1:1:1, followed by a water bath at 37 °C for 30 min. Apple pomace syrup and Greek yogurt were centrifuged at 16,000 rpm for 30 min (Combi 524R, Hanil, Daejeon, Korea), 1 mL of FRAP solution was added in the sample supernatant (50 μL) and reacted for 30 min in a dark place. The absorbance spectrophotometer (UN-1800, Shimadzu, Kyoto, Japan) measured the absorbance at 595 nm. The scavenging ability was calculated from a standard curve using the standard material Trolox (Sigma-Aldrich, Co., St. Louis, MO, USA) for the measured absorbance.

For reducing power, 0.2 M sodium phosphate buffer was adjusted to pH 6.6 according to the method of Oyaizu (1986) [53]. 1% potassium ferricyanide, 10% trichloroacetic acid, and 0.1% ferric chloride were prepared for this experiment. Apple pomace syrup and Greek yogurt were centrifuged at 16,000 rpm for 30 min (Combi 524R, Hanil, Daejeon, Korea). Then 0.2 M sodium phosphate buffer 300 μL and 1% potassium ferricyanide 300 μL was added to the supernatant of the sample (50 μL) and reacted for 20 min in a water bath at 50 °C. Then, 300 μL of 10% trichloroacetic acid and 100 μL of 0.1% ferric chloride were mixed. Absorbance was measured at 700 nm using an absorbance spectrometer (UN-1800, Shimadzu, Japan). The antioxidant value was calculated from the standard curve using a standard material Trolox (Sigma-Aldrich, Co., St. Louis, MO, USA).

### 2.5. Sensory Evaluation

A total of 102 consumers ranging from 20 to 30 years of age (26 males, 76 females) participated in the study. The participants without any food allergies were recruited through the flyers. The participants were asked to refrain from coffee, smoking, or chewing gum at least 1 h before the sensory evaluation. The acceptant test was conducted at the sensory evaluation laboratory in the Prime building of Wonkwang University, Iksan, South Korea. Before the sensory evaluation, participants were informed about the evaluation procedures, rinsing methods, and rating methods. All participants were asked to complete an informed consent form. Participants evaluated the six samples (MGY, APY0.00, APY1.25, APY2.50, APY3.75, APY5.00) for about 50 min at 10:00 a.m. or 3:00 p.m. to avoid lunch times. Each sample (20 g) was coded with random three-digit numbers to prevent response bias. Each sample was presented in a sensory cup (7 cm × 3 cm), and spoons with bottled water for rinsing the mouth.

All samples were provided in a balanced order simultaneously, based on the mutually orthogonal Latin squares (MOLS) design. The participants were asked to evaluate the samples for overall acceptance followed by acceptance of color, flavor, taste with sweet intensity and sour intensity, and aftertaste after swallowing the six samples, using a 9-point hedonic scale anchored on the left ‘dislike extremely’ and on the right ‘like extremely’ (1 = dislike extremely, 5 = average, 9 = like extremely). The evaluation method of purchase intent was evaluated on a 5-point Likert scale (1 = I would never buy, 5 = I would definitely buy).

### 2.6. Statistical Analysis

The results of this experiment were expressed by mean and standard deviation using the XLSTAT program (ver. 2021, Addinsoft, New York, NY, USA). The data from the physicochemical analyses and antioxidant properties were analyzed using analysis of variance (ANOVA), with mean separation using Fisher’s least significant difference (LSD) at a significant level of 0.05. Rating of overall acceptance, color acceptance, flavor acceptance, taste acceptance, sweet intensity, sour intensity, aftertaste acceptance, and purchase intent of each sample were analyzed by analysis of variance (ANOVA), with mean separation using Fisher’s least significant difference (LSD) at a significant level 0.05. Data visualization was achieved using principal component analysis (PCA), and the matrix of 20 variables for six yogurts was analyzed.

## 3. Results and Discussion

### 3.1. Physicochemical Characteristics of Apple Pomace Syrup

The physicochemical characteristics of APS are presented in Table 1. The moisture content of APS was 30.58~39.64%, and the values were significantly higher in syrups fortified with AP compared with those of APS0.00 (*p* < 0.05). These values showed a similar increasing pattern of the moisture content (30.41~45.93%) in syrups produced from six Moroccan date fruit (*Phoenix dactylifera* L.) varieties [54]. AP powder has been reported to have water retention and swelling capacity [55]. The crude ash content of APS was in the range of 0.02~0.21% and showed a high value as the AP content increased, in the order of APS0.00 < APS1.25 < APS2.50 < APS3.75 < APS5.00. APS’s total soluble solids content was 60.20~62.90 °Brix, which is the appropriate standard (over 60%) based on Korea Food Additives Code [56]. However, it is slightly lower than Codex’s description (not less than 70%) of glucose syrup [57]. The pH of APS was 4.28 to 4.77, decreasing significantly as the AP content increased (*p* < 0.05). The results of this study showed a trend similar to those in which the pH decreased with the addition of apple meal to cookies and muffins [24,25]. The decrease in pH could be related to the organic acid content of AP. The titratable acidity of APS was in the range of 0.01 to 0.12%, and it increased significantly as the AP content increased, showing a trend opposite to pH (*p* < 0.05). In the AP meta-analysis results, the malic acid concentration in AP ranged from 0.05 to 3.28 g/100 g [6]. The viscosity of APS was APS1.25 (169.37 cP), APS2.50 (1213.00 cP), APS3.75 (2323.33 cP), and APS5.00 (22606.33 cP), which increased significantly according to the ratio of AP added (*p* < 0.05). Color is one of the important food quality factors in food. When the syrups prepared by adding AP were compared, it was observed that APS with AP (APS1.25, APS2.50, APS3.75, and APS5.00) had lower L* values but higher a* and b* values than those from APS without AP (APS0.00). According to Kim et al. (2002) [58], the total color difference was categorized into 6 groups: imperceptible differences (0.0 to 0.5), slight differences (0.5 to 1.5), just noticeable differences (1.5 to 3.0), marked differences (3.0 to 6.0), extremely marked differences (6.0 to 12.0), and color of a different shade (above 12). The total color difference (ΔE*) of APS increased from 9.10 (APS1.25) to 14.9 (APS5.00). These results demonstrated that coloring by adding AP progressed in a concentration-dependent manner with two groups of extremely marked difference (APS1.25) and color of a different shade (APS2.50, APS3.75, and APS5.00).

### 3.2. Antioxidant Components and Antioxidant Activities of Apple Pomace Syrup

Total polyphenols are phytochemicals, secondary metabolites of plants, widely distributed in plant foods, and are known to have antioxidant, anti-aging, antioxidant, anti-inflammatory, and anti-cancer functions [59]. The antioxidant components, such as total polyphenols and total flavonoids, and antioxidant activities using DPPH radical scavenging activity, ABTS radical scavenging activity, FRAP assay, and reducing power, were presented in Figure 2. The total polyphenol content of APS ranged from 44.34 to 148.48 µg/mL, and it increased significantly as the AP content increased in a concentration-dependent manner. (Figure 2A). According to Antonic et al. (2020) [6], the main polyphenol compounds found in AP were phlorizin, chlorogenic acid, hyperin, epicatechin, quercetin, caffeic acid, and catechin. Especially, phlorizin, a glucoside of phloretin, and a dihydrochalcone are found in most apple cultivars. The total flavonoid content of APS also increased significantly as the AP content increased in a concentration-dependent manner. (Figure 2B). According to Lyu et al. (2020) [2], The major flavonoid compounds of AP contained kaempferol, isorhamnetin, rhamnetin, epicatechin, and quercetin.

The DPPH radical scavenging activity of APS ranged from 0.39 to 83.47%, and it increased significantly according to the addition of AP (*p* < 0.05) (Figure 2C). The ABTS radical scavenging activity of APS also increased significantly according to the AP addition in a concentration-dependent manner (*p* < 0.05) (Figure 2D). According to a previous study [60], the antioxidant activity of ABTS radical scavenging ability was higher in apple pomace (14.15%) than that in apple pulp (5.99%). The FRAP (Ferric reducing Antioxidant Power) assay measures the antioxidant activity of a sample using the principle that Fe^3+^ ions are converted to Fe^2+^ by phenolic compounds at acidic pH [61]. The FRAP increased significantly as the AP content increased (*p* < 0.05) (Figure 2E). The reducing power is also based on an electron transfer reaction; while the FRAP is in an acidic condition, the reducing power is in nearer neutral pH. The reducing power also increased significantly as the AP content increased in neutral pH (*p* < 0.05). (Figure 2F). The increasing antioxidant activities using DPPH radical scavenging activity, ABTS radical scavenging activity, FRAP method, and reducing power might be due to the contents of total polyphenols and total flavonoids.

### 3.3. Physicochemical Characteristics of Greek Yogurt Fortified with Apple Pomace Syrup

The physicochemical properties of Greek yogurt with APS (APY) are presented in Table 2. The moisture content of APY was in the range of 79.51 to 80.06% during 12 h of fermentation. The moisture content increased as the AP-fortified APS was added but decreased as the fermentation continued.

The total soluble solids of APY were significantly decreased in each yogurt sample for the fermentation period, similar to those of the AP-fortified APS, in a concentration-dependent manner. The pH of APY was in the range of 4.28 to 6.85, and the pH of APY0.00 slowly decreased from pH 6.85 at 0 h to pH 4.41 at 12 h as fermentation progressed. Notably, it showed a sharp decrease from 0 to 6 h of the fermentation and a gentle decrease from 9 to 12 h. The pH was significantly decreased in each yogurt sample during the fermentation, and the higher the APS fortification, the faster the pH decreased. The result of the decrease in pH according to the fermentation time was similar to the previous studies of yogurt with mulberry powder [62], Cacao nib powder [63], or black tea [64]. The total organic acid content increases as lactic acid bacteria produce lactic acid during the yogurt fermentation. The titratable acidity of APY was in the range of 0.07~0.52%. The titratable acidity of APY0.00 was 0.07% at 0 h to 0.47% at 12 h, which significantly increased as the fermentation time increased. The higher the APS fortification (APY1.25, APY2.50, APY3.75, and APY5.00) and the longer the fermentation, the titratable acidity significantly increased (*p* < 0.05). According to Ahmad et al. (2020) [33], fortification of apple peel extract in yogurt ice cream decreased the pH, increased the acidity, and had higher viable counts of probiotics *(Lactobacillus acidophilus* and *Bifidobacterium lactis*) compared to the control samples.

In the chromaticity measurement results of APY, the L* value of APY0.00 was increased from 81.65 to 87.95 during 12 h of fermentation, and the trend was similarly shown in all yogurt samples. As the AP content increased, the L* value decreased in a concentration-dependent manner. Whereas the longer the fermentation period, the L* value increased. Under the same fermentation time duration, a* value of AP-fortified APY (APY1.25, APY2.50, APY3.75, and APY5.00) increased as the AP fortification increased compared to the APY0.00. The b* value of APY0.00 was increased from 7.35 to 11.73 during 12 h of fermentation, whereas the b* value of AP fortified APY showed a significant decrease during the fermentation period. Under the same fermentation period, the b* value increased significantly as the fortification of AP in yogurt increased. The total color difference (ΔE*) of APY increased as the AP fortification increased compared to the APY0.00 under the same fermentation period. In contrast, the ΔE* of AP-fortified APY (APY1.25, APY2.50, APY3.75, and APY5.00) decreased in value during fermentation. These results demonstrated that the total color difference of APY was categorized into three groups as marked differences (at 3 hr of fermentation, APY0.00), the color of a different shade (at 0 hr of fermentation, APY1.25, APY2.50, APY3.75, and APY5.00), and extremely marked differences (the others from marked differences and color of a different shade). In apples, several classes of polyphenols, such as phenolic acids, anthocyanins, and flavonoids, are responsible for the color of apple skins, with color changes during the processing stages. According to Fanyuk et al. (2022) [65], preharvest phenylalanine treatment for the red coloration of apples increased the red coloring intensity with total flavonoids, anthocyanin contents, and antioxidant activity compared with the control fruit.

### 3.4. Antioxidant Contents and Antioxidant Activities of Greek Yogurt Fortified with Apple Pomace Syrup

The antioxidant contents and activities of Greek yogurt with APS (APY) are presented in Figure 3. The total polyphenol content rapidly increased in all yogurt samples until 6 h of fermentation but changed slowly thereafter (Figure 3A). APY5.00 showed the highest total polyphenol content than the other yogurts as fermentation continued and AP fortification increased, followed by APY3.75 > APY2.50 > APY1.25 > APY0.00. As shown in Figure 2A, the polyphenol content of APY significantly increased in an AP concentration-dependent manner. This was similar to the report in a previous study [64] that the total polyphenol content of black tea Greek yogurt significantly increased up to the 6-h fermented group. According to Santos et al. (2017) [66], the total polyphenol content increased in goat milk yogurt fortified with grape pomace extract with a protective effect on the viability of *L. acidophilus*. According to Jovanovic et al. (2020) [32], yogurt fortified with AP flour (0% to 5%), the addition of AP flour increased the total polyphenol content of yogurts in a concentration-dependent manner. In Figure 3B, the DPPH radical scavenging activity of Greek yogurt with APS increased in a concentration-dependent manner, and this trend continued during fermentation. It can be observed that the fortification of APS increases the total polyphenol contents and DPPH radical scavenging activity of Greek yogurt in a concentration-dependent manner. The ABTS radical scavenging activity of APY showed a similar trend to the total polyphenol contents and DPPH radical scavenging activity (Figure 3C). The antioxidant activities of FRAP assay and reducing power, the reduction of ferric iron (Fe^3+^) to ferrous iron (Fe^2+^) monitored under acidic pH and neutral pH, increased significantly during fermentation time (*p* < 0.05) (Figure 3D,E). The FRAP of fermented milk with green tea powder was significantly higher in the fermented group than in the non-fermented group [67], which showed comparable results in this study. According to Jovanovic et al. (2020) [32], strong correlation coefficients between total polyphenol contents and DPPH and FRAP activities showed in the supernatant obtained from the yogurt fortified with AP flour. The main polyphenol compounds, such as phlorizin, catechin, epicatechin, chlorogenic acid, caffeic acid, hyperin, and quercetin, were found in apple pomace [2,6,68].

### 3.5. Consumer Acceptance of Greek Yogurt Fortified with Apple Pomace Syrup

Sensory evaluation is one of the critical food quality factors in consumers’ food choices. In this study, a consumer acceptance test was performed to evaluate the acceptability of the APY. The results of consumer acceptance and purchase content are presented in Figure 4. The overall acceptance followed by acceptance of color, flavor, taste with sweet and sour intensity, and aftertaste after swallowing was evaluated for a total of 6 samples (MGY, APY0.00, APY1.25, APY2.50, APY3.75, APY5.00) with 12 h fermentation. The MGY (commercial Greek yogurt) had the lowest overall acceptance, whereas the highest in color acceptance, sour intensity, and creamy acceptance. APY1.25 showed significantly higher overall acceptance, taste acceptance, sweet intensity, aftertaste acceptance, and purchase intent (*p* < 0.05). The color acceptance, sour intensity, and creamy acceptance of Greek yogurt fortified with APS had significantly decreased with the fortification of AP, whereas flavor acceptance was not significant in all samples (*p* < 0.05). In this result, the consumer acceptance test suggests that Greek yogurt fortified with APS1.25 is considered the most acceptable when manufactured. In the past study of yogurt fortified with 1%, 3%, and 5% AP flour, the highest score of the acceptable sensory quality of yogurt fortified with AP flour was the sample with the addition of 3% AP flour, even though the most prominent antioxidant activity and cytotoxic activity against colon cancer were related to the yogurt with 5% [32].

The data were visualized using PCA to examine the correlations between physicochemical characteristics and antioxidant activities with consumer acceptance toward the Greek yogurt fortified with APS (Figure 5). The scores plot defined by the first two principal components (PC1 and PC2) account for 69.11 and 13.25% of the variability and showed a clear separation of APS samples. APS1.25 assumed highly positive PC1 and PC2 values in the I quadrant of the plot, characterized by overall acceptance, purchase intent, taste acceptance, and sweet intensity. Three Greek yogurts such as APY2.50, APY3.75, and APY5.00, located in the IV quarter, are characterized by color values (a* value and b* value) and antioxidant activities (DPPH, ABTS, FRAP, and RP). MGY is a commercial sample located in the III quarter, characterized by creamy acceptance and moisture acceptance, assuming the lowest PC1 value. APY0.00 is a control sample located in the II quarter, characterized by aftertaste acceptance, flavor acceptance, color acceptance, and sour intensity. APY1.25, located in the I quadrant of the PCA, scored the highest in overall acceptance, taste acceptance, aftertaste acceptance, and purchase intent (Figure 4). Whereas three APY (APY2.50, APY3.75, APY5.00), located in the IV quarter of the PCA, scored the lowest in overall acceptance, color acceptance, taste acceptance, sour intensity, aftertaste acceptance, and purchase intent (Figure 4).

## 4. Conclusions

Greek yogurt fortified with apple pomace syrup (APS), prepared by varying concentrations of apple pomace (AP), was developed. The physicochemical characteristics and antioxidant activities of the yogurt were related to the amount of APS added in a concentration-dependent manner. Greek yogurt fortified with APS (APY) showed increased total polyphenols and antioxidant activities as AP content and fermentation time increased. However, an investigation of consumer acceptance of Greek yogurt fortified with APS (APY) revealed that the fortification of 1.25% APS represents the optimal amount. The APY1.25 showed the most desirable consumer overall acceptance with sweet intensity and the best purchase intent and proved crucial in formulating and accepting new products. The presented results broaden our knowledge of the relationship between consumer perception and acceptance and the important attributes of Greek yogurt fortified with APS, which can be a key point in developing yogurts that satisfy consumer needs or wants.

## Figures and Tables

**Figure 1 foods-12-01856-f001:**
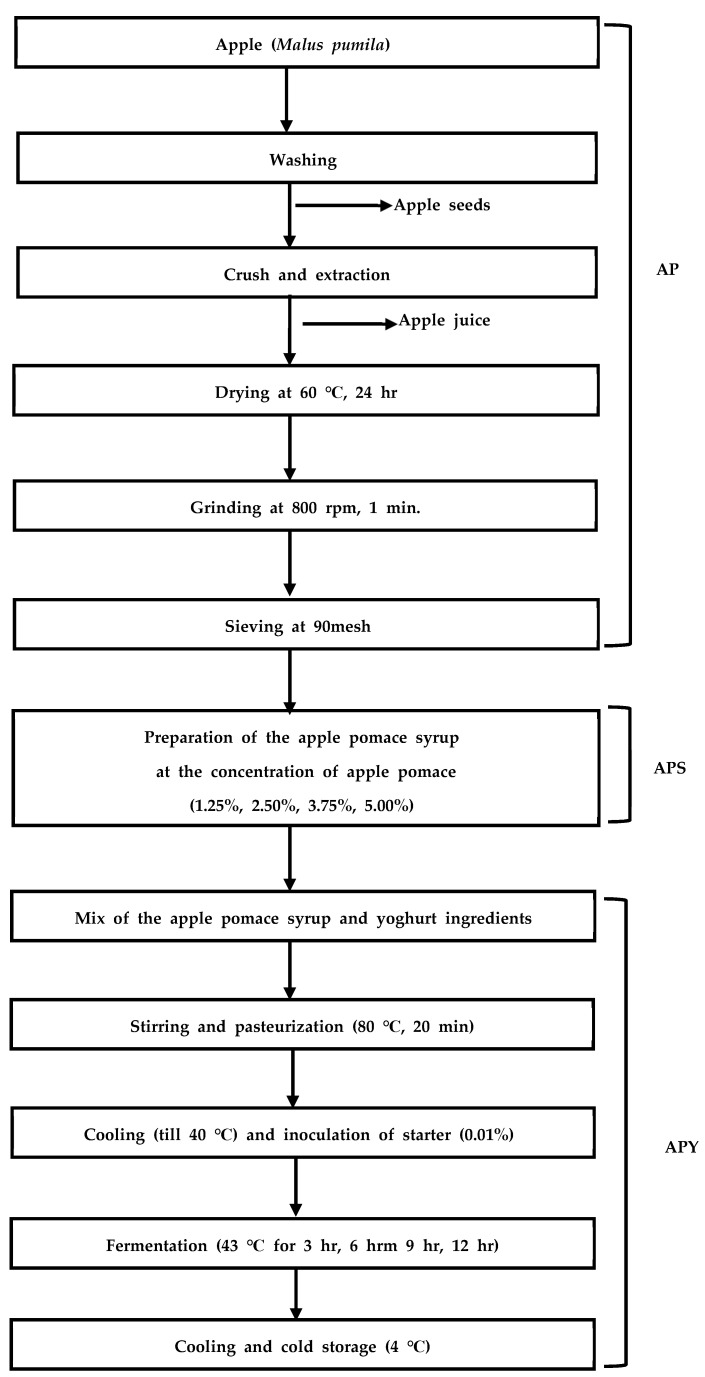
Preparation of Greek yogurt fortified with APS.

**Figure 2 foods-12-01856-f002:**
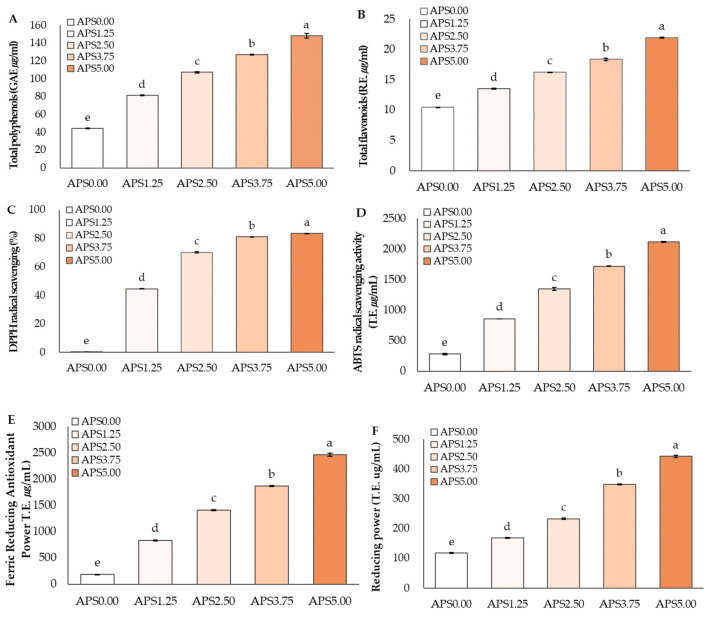
Antioxidant contents (**A**,**B**) and antioxidant activities (**C**–**F**) of APS. APS0.00, APS1.25, APS2.50, APS3.75, APS5.00: apple pomace syrups (APS) prepared by ample pomace (AP) powder with different addition concentrations (0.00 to 5.00%); Mean values with different letters are significantly different at the *p* < 0.05.

**Figure 3 foods-12-01856-f003:**
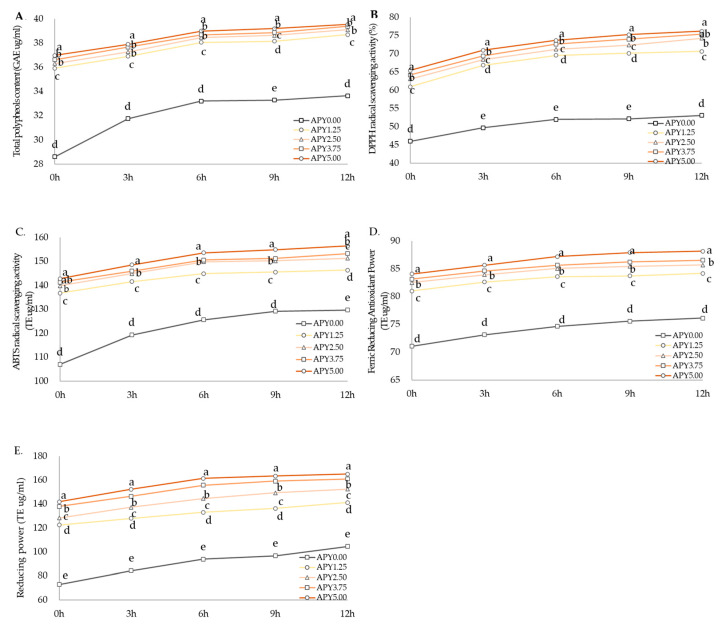
Total polyphenols (**A**) and antioxidant activities (**B**–**E**) of Greek yogurt fortified with APS. Mean values in each fermentation time (a–e) with different superscript letters are significantly different at *p*-value < 0.05.

**Figure 4 foods-12-01856-f004:**
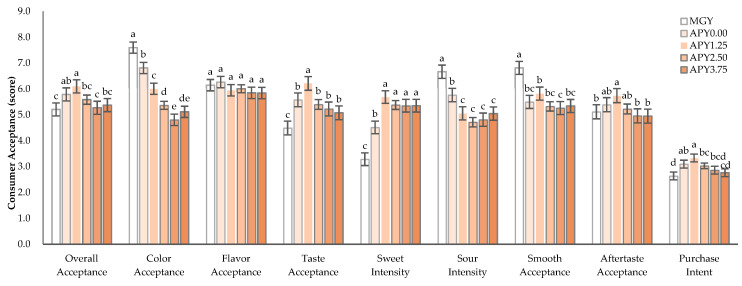
Consumer acceptance of Greek yogurt fortified with APS. MGY: prepared by commercial Greek yogurt, and APY0.00, APY1.25, APY2.50, APY3.75, APY5.00: Greek yogurt prepared by APS from AP powder with different addition concentrations (0.00 to 5.00%); Mean values with different letters are significantly different at *p*-value < 0.05.

**Figure 5 foods-12-01856-f005:**
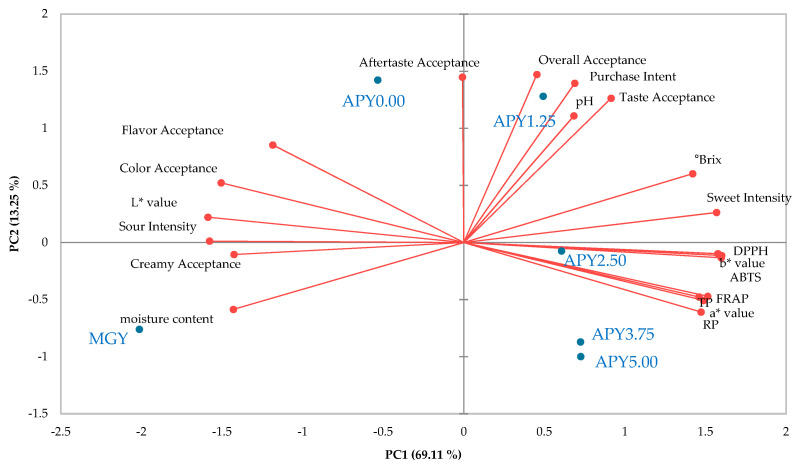
Biplot representation of the principal component analysis of physicochemical, antioxidant activities, and consumer acceptance data by 102 consumers for Greek yogurt fortified with APS.

**Table 1 foods-12-01856-t001:** Physicochemical characteristics of APS.

		APS0.00	APS1.25	APS2.50	APS3.75	APS5.00
Moisture content (%)		30.58 ± 0.28 ^b^	35.41 ± 0.28 ^a^	35.59 ± 0.16 ^a^	38.20 ± 0.60 ^a^	39.64 ± 0.58 ^a^
Crude ash content (%)		0.02 ± 0.02 ^c^	0.03 ± 0.03 ^c^	0.14 ± 0.01 ^b^	0.16 ± 0.02 ^b^	0.21 ± 0.03 ^a^
Total soluble solids (°Brix)		61.37 ± 0.06 ^d^	62.03 ± 0.06 ^c^	62.90 ± 0.00 ^a^	60.20 ± 0.01 ^e^	62.23 ± 0.06 ^b^
pH		4.77 ± 0.01 ^a^	4.61 ± 0.00 ^b^	4.46 ± 0.01 ^c^	4.32 ± 0.01 ^d^	4.28 ± 0.01 ^e^
Titratable acidity (%)		0.01 ± 0.00 ^e^	0.03 ± 0.00 ^d^	0.06 ± 0.00 ^c^	0.08 ± 0.00 ^b^	0.12 ± 0.00 ^a^
Viscosity (cP)		ND	169.37 ± 3.06 ^b^	1213.00 ± 6.93 ^b^	2323.33 ± 2.08 ^b^	33606.33 ± 25.40 ^a^
Color	L* value	60.10 ± 0.69 ^a^	54.10 ± 0.17 ^b^	50.23 ± 0.32 ^c^	48.67 ± 0.32 ^d^	44.27 ± 0.84 ^e^
	a* value	−1.00 ± 0.00 ^d^	0.67 ± 0.06 ^c^	1.97 ± 0.15 ^b^	2.53 ± 0.29 ^a^	2.37 ± 0.12 ^a^
	b* value	−2.03 ± 0.31 ^d^	6.03 ± 0.40 ^b^	7.27 ± 0.06 ^a^	6.80 ± 0.35 ^a^	2.77 ± 0.35 ^e^
	ΔE*	-	9.10 ± 0.42 ^d^	12.50 ± 0.22 ^c^	13.30 ± 0.23 ^b^	14.9 ± 0.70 ^a^

APS0.00, APS1.25, APS2.50, APS3.75, APS5.00: apple pomace syrups prepared from ample pomace (AP) powder with different addition concentration (0.00 to 5.00%); Means ± standard deviations of triplicate samples (*n* = 3); ^a–e^ Mean values within a row with different superscript letters are significantly different at the *p* < 0.05.; ND: not detected.

**Table 2 foods-12-01856-t002:** Physicochemical characteristics of Greek yogurt fortified with APS.

		Fermentation Time (hr)	APY0.00	APY1.25	APY2.50	APY3.75	APY5.00
Moisture content (%)		0	79.90 ± 0.06 ^aA^	79.91 ± 0.01 ^aA^	79.92 ± 0.02 ^aA^	79.96 ± 0.05 ^aA^	80.06 ± 0.06 ^aA^
		3	79.81 ± 0.12 ^aB^	79.83 ± 0.03 ^aB^	79.89 ± 0.05 ^aAB^	79.93 ± 0.05 ^aAB^	80.02 ± 0.24 ^aA^
		6	79.55 ± 0.15 ^bB^	79.61 ± 0.09^bAB^	79.65 ± 0.04 ^bAB^	79.67 ± 0.02 ^bAB^	79.70 ± 0.03 ^bA^
		9	79.52 ± 0.07 ^bB^	79.54 ± 0.08 ^bB^	79.58 ± 0.06 ^cAB^	79.61 ± 0.03 ^cAB^	79.66 ± 0.13 ^cA^
		12	79.51 ± 0.07 ^bC^	79.53 ± 0.08 ^bBC^	79.58 ± 0.05 ^cABC^	79.61 ± 0.02 ^cAB^	79.66 ± 0.03 ^cA^
Crude ash content (%)		0	1.50 ± 0.04 ^bB^	1.52 ±0.02 ^cAB^	1.53 ± 0.02 ^cAB^	1.53 ± 0.03 ^cAB^	1.54 ± 0.00 ^cA^
		3	1.51 ± 0.03 ^bB^	1.53 ± 0.01 ^cAB^	1.54 ± 0.01 ^cA^	1.55 ± 0.01 ^cA^	1.55 ± 0.01 ^cA^
		6	1.55 ± 0.01 ^abC^	1.57 ± 0.01 ^bBC^	1.57 ± 0.01 ^bAB^	1.58 ± 0.02 ^bAB^	1.59 ± 0.01 ^bA^
		9	1.59 ± 0.05 ^aA^	1.60 ± 0.03 ^aA^	1.61 ± 0.02 ^aA^	1.62 ± 0.01 ^aA^	1.63 ± 0.02 ^aA^
		12	1.60 ± 0.01 ^aD^	1.61 ± 0.02 ^aCD^	1.62 ± 0.01 ^aBC^	1.63 ± 0.01 ^aAB^	1.64 ± 0.01 ^aA^
Total soluble solids (°Brix)		0	22.20 ± 0.00 ^aA^	21.70 ± 0.00 ^aB^	21.70 ± 0.00 ^aB^	21.70 ± 0.00 ^aB^	21.23 ± 0.05 ^aC^
		3	20.70 ± 0.20 ^bA^	20.65 ± 0.10 ^bA^	20.65 ± 0.06 ^bA^	20.60 ± 0.00 ^bA^	20.60 ± 0.00 ^bA^
		6	20.40 ± 0.00 ^cA^	20.00 ± 0.00 ^cB^	20.00 ± 0.00 ^cB^	20.00 ± 0.00 ^cB^	19.90 ± 0.08 ^cC^
		9	20.35 ± 0.06 ^cA^	20.00 ± 0.00 ^cB^	20.00 ± 0.00 ^cB^	20.00 ± 0.00 ^cB^	19.88 ± 0.10 ^cC^
		12	19.58 ± 0.26 ^dA^	19.40 ± 0.14 ^dAB^	19.30 ± 0.12 ^dB^	19.88 ± 0.10 ^dBC^	19.00 ± 0.00 ^dC^
Reducing sugar (mg/mL)		0	0.55 ± 0.00 ^dC^	0.66 ± 0.01 ^dB^	0.66 ± 0.01 ^cB^	0.68 ± 0.01 ^dA^	0.68 ± 0.01 ^dA^
		3	0.56 ± 0.00 ^cC^	0.68 ± 0.00 ^cB^	0.68 ± 0.00 ^bB^	0.69 ± 0.01 ^cB^	0.70 ± 0.01 ^cA^
		6	0.57 ± 0.00 ^bcD^	0.69 ± 0.01 ^bcC^	0.69 ± 0.00 ^bD^	0.70 ± 0.00 ^bcB^	0.71 ± 0.00 ^bcA^
		9	0.57 ± 0.00 ^abE^	0.69 ± 0.00 ^abD^	0.70 ± 0.00 ^aC^	0.70 ± 0.00 ^abB^	0.71 ± 0.00 ^abA^
		12	0.58 ± 0.00 ^aD^	0.70 ± 0.00 ^aC^	0.70 ± 0.00 ^aB^	0.71 ± 0.00 ^aAB^	0.71 ± 0.00^aA^
pH		0	6.85 ± 0.01 ^aA^	6.74 ± 0.01 ^aB^	6.71 ± 0.02 ^aB^	6.67 ± 0.03 ^aC^	6.66 ± 0.03 ^aC^
		3	5.38 ± 0.06 ^bA^	5.12 ± 0.04 ^bB^	5.12 ± 0.06 ^bB^	5.10 ± 0.05 ^bB^	5.08 ± 0.03 ^bB^
		6	4.77 ± 0.64 ^cA^	4.56 ± 0.01 ^cB^	4.56 ± 0.02 ^cB^	4.56 ± 0.02 ^cB^	4.56 ± 0.01 ^cB^
		9	4.46 ± 0.06 ^dA^	4.38 ± 0.01 ^dB^	4.37 ± 0.00 ^dB^	4.37 ± 0.02 ^dB^	4.36 ± 0.01 ^dB^
		12	4.41 ± 0.03 ^eA^	4.30 ± 0.02 ^eB^	4.28 ± 0.01 ^eB^	4.28 ± 0.01 ^eB^	4.28 ± 0.02 ^eB^
Titratable acidity (%)		0	0.07 ± 0.00 ^eC^	0.07 ± 0.00 ^eC^	0.07 ± 0.00 ^eC^	0.07 ± 0.00 ^eB^	0.09 ± 0.00 ^eA^
		3	0.39 ± 0.01 ^dC^	0.40 ± 0.00 ^dC^	0.41 ± 0.01 ^dB^	0.42 ± 0.01 ^dB^	0.43 ± 0.00 ^dA^
		6	0.41 ± 0.00 ^cE^	0.45 ± 0.00 ^cD^	0.46 ± 0.00 ^cC^	0.46 ± 0.00 ^cB^	0.47 ± 0.00 ^cA^
		9	0.45 ± 0.00 ^bD^	0.48 ± 0.00 ^bC^	0.49 ± 0.00 ^bB^	0.49 ± 0.00 ^bB^	0.51 ± 0.00 ^bA^
		12	0.47 ± 0.00 ^aD^	0.50 ± 0.00 ^aC^	0.51 ± 0.00 ^aB^	0.52 ± 0.00 ^aB^	0.52 ± 0.00 ^aA^
Color	L* value	0	81.65 ± 0.10 ^cA^	74.48 ± 0.49 ^cB^	69.60 ± 0.08 ^dC^	68.58 ± 0.10 ^cD^	67.48 ± 0.53 ^cE^
		3	85.03 ± 1.15 ^bA^	81.40 ± 0.39 ^bB^	79.40 ± 0.35 ^cC^	78.18 ± 0.78 ^cD^	77.40 ± 0.44 ^bD^
		6	87.83 ± 0.29 ^aA^	84.70 ± 0.12 ^aB^	83.80 ± 0.14 ^bC^	82.85 ± 0.24 ^bD^	82.53 ± 0.05 ^aE^
		9	87.90 ± 0.18 ^aA^	84.73 ± 0.13 ^aB^	83.75 ± 0.06 ^aC^	82.70 ± 0.28 ^aD^	82.68 ± 0.05 ^aD^
		12	87.95 ± 0.13 ^aA^	84.98 ± 0.10 ^aB^	84.10 ± 0.14 ^aC^	83.15 ± 0.13 ^aD^	82.88 ± 0.01 ^aE^
	a* value	0	−4.70 ± 0.00 ^dC^	1.15 ± 0.10 ^dB^	1.20 ± 0.18 ^dB^	1.75 ± 0.17 ^cA^	1.93 ± 0.54 ^bA^
		3	−2.43 ± 0.21 ^cC^	1.40 ± 0.08 ^cB^	1.50 ± 0.08 ^cB^	1.78 ± 0.10 ^cA^	1.80 ± 0.08 ^bA^
		6	−1.48 ± 0.17 ^bD^	1.45 ± 0.06 ^cC^	1.80 ± 0.08 ^bB^	2.00 ± 0.08 ^bA^	2.08 ± 0.10 ^bA^
		9	−1.33 ± 0.10 ^bE^	1.68 ± 0.05 ^bD^	2.25 ± 0.06 ^aC^	2.53 ± 0.10 ^aB^	2.80 ± 0.08 ^aA^
		12	−1.10 ± 0.08 ^aE^	1.83 ± 0.10 ^aD^	2.38 ± 0.05 ^aC^	2.60 ± 0.08 ^aB^	2.90 ± 0.08 ^aA^
	b* value	0	7.35 ± 0.13 ^dE^	15.88 ± 0.10 ^aD^	16.85 ± 0.17 ^aC^	17.60 ± 0.36 ^aB^	18.13 ± 0.22 ^aA^
		3	10.35 ± 0.17 ^cE^	14.28 ± 0.13 ^bD^	14.70 ± 0.29 ^bC^	15.18 ± 0.26 ^bB^	15.53 ± 0.15 ^bA^
		6	11.15 ± 0.19 ^bD^	14.00 ± 0.00 ^cC^	14.15 ± 0.06 ^cC^	14.75 ± 0.17 ^cB^	15.03 ± 0.10 ^cA^
		9	11.25 ± 0.17 ^bB^	13.08 ± 0.15 ^dC^	14.15 ± 0.10 ^cB^	14.23 ± 0.05 ^dB^	14.63 ± 0.05 ^dA^
		12	11.73 ± 0.10 ^aD^	12.75 ± 0.19 ^eC^	13.98 ± 0.05 ^cB^	14.15 ± 0.13 ^dB^	14.50 ± 0.08 ^dA^
	ΔE*	0	-	12.6 ± 0.32 ^dA^	16.4 ± 0.18 ^cA^	17.8 ± 0.29 ^bA^	19.1 ± 0.59 ^aA^
		3	5.0 ± 0.92 ^c**B**^	9.2 ± 0.14 ^b**B**^	9.9 ± 0.30 ^bBC^	10.7 ± 0.29 ^aB^	11.3 ± 0.26 ^aB^
		6	7.9 ± 0.21 ^dA^	9.5 ± 0.05 ^cC^	9.6 ± 0.06 ^cC^	10.0 ± 0.12 ^bC^	10.2 ± 0.13 ^aC^
		9	8.0 ± 0.11 ^dA^	9.1 ± 0.14 ^cC^	9.9 ± 0.10 ^bB^	10.0 ± 0.10 ^bC^	10,5 ± 0.07 ^aC^
		12	8.4 ± 0.11 ^dA^	9.1 ± 0.17 ^cC^	10.0 ± 0.04 ^bB^	10.0 ± 0.05 ^bC^	10.5 ± 0.09 ^aC^

APY0.00, APY1.25, APY2.50, APY3.75, APY5.00: Greek yogurt prepared by apple pomace syrups (APS) from ample pomace (AP) powder with different addition concentrations (0.00 to 5.00%); Means ± standard deviations of triplicate samples (*n* = 3); Mean values within each column (^a–e^) and each row (^A~E^) with different superscript letters are significantly different at the *p* < 0.05.

## Data Availability

The data used to support the findings of this study can be made available by the corresponding author upon request.

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
