# Peer review of "Physicochemical Characteristics, Antioxidant Properties and Consumer Acceptance of Greek Yogurt Fortified with Apple Pomace Syrup"

_foods, 2023, doi:10.3390/foods12091856_

Round 1

Reviewer 1 Report

In the presented study, the authors have evaluated Greek yogurt fortified with apple pomace syrup regarding physicochemical characteristics and antioxidant properties and consumer acceptance test. It is a very promising idea. I enjoyed reading the article. However, some minor corrections are necessary.

Specific comments:

Page 2, at the end of the introduction section, highlight the novelty of the research.

Page 2, line 68, is there any particular reason why this kind of apples was selected for the research?

Page 2, line 82, provide the initial and final moisture content of the apple pomace.

Page 7, Table 1, calculate ΔE for apple pomace syrup color values

Page 10, Table 2, calculate ΔE for Greek yogurt fortified with apple pomace syrup color values

Page 13, include standard deviation in Figure 4

Page 13, discuss in more detail the PCA biplot

Minor editing of English language required.

Author Response

Thank you so much for the review. Please see the attachment.

Reviewer 2 Report

Detailed recommendation:

Please choose type of article

Abstract: in my opinion you should add more data to the abstract section.

Key words: please add: functional food

Please calculate ∆E in color parameters.

Introduction: give more information about functional food and its pro-health properties of bioactive substance which are in apple pomaces.

 Minor editing of English language required

Author Response

(The authors gave the same response as above.)

Reviewer 3 Report

The paper ‘Physicochemical Characteristics, Antioxidant Properties and Consumer Acceptance of Greek Yogurt Fortified with Apple Pomace Syrup’ by Kim et al. discussed the physicochemical characteristics and antioxidant properties of Greek yogurt containing various concentrations of apple pomace syrup (APS)  (APS0.00, APS1.25, APS2.50, APS3.75, 60 APS5.00). Consumer acceptance test with Greek yogurt fortified with APS (APY) was also conducted, as an attempt to find a positive application for AP by developing functional yogurt fortified with APS. The work quality is good in general, however, the following points should be addressed carefully:

1.       More details on the used starter cultures are needed.

2.       Line 94, don’t start the sentences with numbers, apply this comment to the whole text.

3.       Why you fermented the yogurt at different time intervals? It should be pH-based not time-based.

4.       There are different literature in the last few years covering the same topic, could you clearly mention the difference between your paper and the following papers and include them in the introduction? Then mention the novelty of your work.

1. Wang, X., Kristo, E., & LaPointe, G. (2020). Adding apple pomace as a functional ingredient in stirred-type yogurt and yogurt drinks. Food Hydrocolloids100, 105453.

2. Wang, X., Kristo, E., & LaPointe, G. (2019). The effect of apple pomace on the texture, rheology and microstructure of set type yogurt. Food Hydrocolloids91, 83-91.

3. Jovanović, M., Petrović, M., Miočinović, J., Zlatanović, S., Laličić Petronijević, J., Mitić-Ćulafić, D., & Gorjanović, S. (2020). Bioactivity and sensory properties of probiotic yogurt fortified with apple pomace flour. Foods9(6), 763.

4. Popescu, L., Ceșco, T., Gurev, A., Ghendov-Mosanu, A., Sturza, R., & Tarna, R. (2022). Impact of apple pomace powder on the bioactivity, and the sensory and textural characteristics of

5.       Figure 1 can be designed in a better way by including some pictures of the syrup and the end product to have an idea of what it looks like. Also there are some faults in the text; e.g. in the fermentation section what 6hrm mean?

6.       Try to avoid old references, e.g. DPPH and ABTS parts, the following can be used instead: Garavand, F., Daly, D. F., & Gomez-Mascaraque, L. G. (2022). Biofunctional, structural, and tribological attributes of GABA-enriched probiotic yoghurts containing Lacticaseibacillus paracasei alone or in combination with prebiotics. International Dairy Journal129, 105348.

7.       What not detected mean for the viscosity in Table 1?

8.       Significant difference letters are missed in Figure 2 C and F.

9.       Table 2, how is moisture content going to change during the fermentation?

Author Response

(The authors gave the same response as above.)
